

# Antecedent wetness and rainfall information in landslide threshold definition

Binru Zhao[1,2], Qiang Dai[3,2], Dawei Han[2], Huichao Dai[1], Jingqiao Mao[1], Lu Zhuo[2]

[1]College of Water Conservancy and Hydropower Engineering, Hohai University, Nanjing, China.
[2]Department of Civil Engineering, University of Bristol, Bristol, UK.
[3]Key Laboratory of VGE of Ministry of Education, Nanjing Normal University, Nanjing, China.

*Correspondence to*: Qiang Dai (bz17336@bristol.ac.uk)

**Abstract.** For rainfall-induced landslides, their occurrence is attributed to both the antecedent wetness condition and the recent rainfall condition. However, when defining rainfall thresholds for the landslide occurrence, these two types of

information have been used incompletely or implicitly, which may affect the threshold's predictive capability. This study aims to investigate how to make a better use of these two types of information in the landslide threshold definition. Here four types of thresholds are proposed, by including different variables that are responsible for landslide occurrences, these thresholds could represent different cases, like whether to include the antecedent wetness information or whether to consider the recent rainfall condition explicitly. The predictive capability of these thresholds is then compared crossly with the help of

the receiver operating characteristic (ROC) approach. We carry out this study in a northern Italian region called Emilia-Romagna. Results show that the antecedent wetness condition plays a crucial role in the occurrence of rainfall-induced landsides. It is beneficial for the threshold's predictive capability to explicitly include the antecedent wetness information and the recent rainfall in the definition of landslide thresholds. When including soil moisture information in landslide threshold, the reliability of the soil moisture measurement is a key factor affecting the threshold's prediction performance.

**1 Introduction**

Landslides are one of the most frequent and widespread natural hazards, posing threat to human lives and local infrastructures. These threats increase with the continuous development in the mountain areas. Landslide alerts or early warnings are able to provide useful information for disaster managers and emergency planners to mitigate the related socio-economic risk (Keefer et al., 1987;Jakob et al., 2006;Mirus et al., 2018).

The occurrence of landslides is a result of multiple factors, such as rainfall, snowmelt, earthquakes, human activities, etc. Among these factors, rainfall is the most common driving force. Rainfall-induced landslides are typically due to the increase of the negative pore-water pressure which decreases the shear strength of the soil and leads to the slope failures. This type of landslide usually follows a long period of the wet condition and then triggered by intense rainfall. Given rainfall could be

seen as a good proxy for both the antecedent wetness condition and the recent rainfall condition, it is widely used to derive



the threshold for landslide occurrence based on an empirical approach. Generally, two features of the rainfall event are identified and labelled with landslide occurrence or non-occurrence. Hereafter, a line or zone is derived to separate rainfall events inducing landslides from those without landslide hazards. The separation line or zone can be determined visually (Caine, 1980) or by some statistical methods, like the method based on Bayesian inference (Guzzetti et al., 2007a;Guzzetti et

al., 2007b) and the frequentist approach (Brunetti et al., 2010). The most common variables used to characterize rainfall events are rainfall intensity-duration (ID) and cumulated event rainfall-rainfall duration (ED). Various rainfall thresholds for landslide occurrences have been proposed and applied (Peruccacci et al., 2012;Segoni et al., 2014;Gariano et al., 2015;Peruccacci et al., 2017;Guzzetti et al., 2007a;Guzzetti et al., 2007b). Although these thresholds are the main tool in landslide early warning systems, their shortcomings are frequently recognized and discussed. For example, the information

of the antecedent wetness or the recent rainfall is not explicitly considered in setting the threshold. When deriving rainfall thresholds, the rainfall events responsible for landslides have a duration ranging from one day to a few months. For rainfall events with short durations, they are likely to neglect the information of the antecedent wetness. As for the rainfall event with long durations, although it implicitly includes the antecedent wetness information, it is not able to reflect the real causal relationship between rainfall events and landslides, because in this case, there may be an intensity peak, which is the real

trigger of landslides, preceded by a rainfall period which predisposes the slope to failure (Bogaard and Greco, 2018). However, the intensity calculated based on such a long period flattens the intensity peak, ignoring the role of the rainfall trigger.

To more explicitly take into account the antecedent wetness condition and the recent rainfall, several attempts have been

proposed to derive the hydro-meteorological thresholds, which are based on the concept that the landslide occurrence is attributed to both the antecedent wetness condition (hydrological information) and the final rainfall trigger (meteorological information). They incorporate measures of the antecedent wetness condition into the definition of thresholds. In some landslide early warning systems, the antecedent cumulated rainfall over a certain period is calculated to characterize the antecedent wetness condition, which is used together with the recent rainfall amounts to derive the thresholds. For example,

Chleborad et al. (2008) and Scheevel et al. (2017) made use of the recent 3-day rainfall and the antecedent 15-day rainfall to define the threshold, while Lee and Park (2015) considered the recent daily rainfall and the antecedent 3-day rainfall information. Besides the antecedent cumulated rainfall, Glade et al. (2000) employed an Antecedent Precipitation Index (API) to describe the antecedent wetness condition, which could take the loss of the antecedent rainfall into consideration. In addition to the use of rainfall information, some direct measures or proxies for the antecedent wetness condition were also

explored (Crozier, 1999;Godt et al., 2006;Ponziani et al., 2012;Gabet et al., 2004). Mirus et al. (2018) accounted for the antecedent wetness condition with direct subsurface hydrological measurements, which are then combined with the rainfall information to define the threshold for landslides. The derived thresholds show improved performances in landslide alert systems. The catchment storage is also regarded as a source of information on the antecedent wetness condition. Ciavolella et al. (2016) included the catchment storage in the definition of landslide thresholds in a catchment in the northern



Apennines (Italy). The hydro-meteorological threshold based on event rainfall and catchment specific storage performs slightly better than the rainfall intensity-duration threshold. Segoni et al. (2018) substituted the antecedent rainfall accumulated over long periods with the soil moisture thresholds in the rainfall thresholds of the regional-scale landslide early warning system. A back analysis demonstrated this approach is able to reduce false alarms and missed alarms. These

examples all explicitly consider the antecedent wetness condition and the recent rainfall when defining the thresholds for rainfall-induced landslides. There are other studies integrating the antecedent wetness condition and the recent rainfall into one variable. Zhuo et al. (2019) used the remotely sensed soil moisture prior to landslides to include these two types of information and proposed the soil moisture thresholds for landslides under different environmental conditions (land cover, soil type and type). The thresholds proposed by the published studies consider the antecedent wetness condition and (or) the

recent rainfall condition implicitly or explicitly. However, to the authors' knowledge, such studies lack a more thorough analysis of the role the antecedent wetness condition and the recent rainfall in the landslide occurrence and their usage in the threshold definition, though their importance is stressed in a series of works (Ciavolella et al., 2016;Bogaard and Greco, 2018).

Therefore, this study aims to explore how to make a better use of the antecedent wetness information and the recent rainfall information in the definition of landslide thresholds. First, we intend to investigate the effect of incorporating antecedent wetness information to the landslide threshold. Although the role of the antecedent wetness condition is often regarded to be of great importance in the landslide occurrence, its direct impact on the threshold performance is rarely investigated. Second, we attempt to answer the question whether it is necessary to explicitly consider the antecedent wetness condition and the

recent rainfall when defining thresholds for landslides? In order to address these two issues, four types of thresholds are proposed, by including different variables that are responsible for landslide occurrences, these thresholds could represent different cases, like whether to include the antecedent wetness information or whether to consider the recent rainfall condition explicitly. The predictive capability of these thresholds is compared crossly with the help of the receiver operating characteristic (ROC) approach.  Here the wetness condition is characterized by the Antecedent Precipitation Index (API) due

to its simple formulation and low requirement for data. We carry out this study in a northern Italian region called Emilia-Romagna, where the landslide records and hydrometeorological data are abundant and available.

This paper is organized as follows. Section 2 introduces the study area and data sources. Section 3 details the methods used in this study. The results are described in Section 4, followed by further discussions and limitations in Section 5. In the final

section, we outlined the conclusions and future works.



## 2 Study Area and Data Sources

### 2.1 Study Area

Emilia-Romagna region is located in the north of Italy and is one of the most fertile and productive regions in the country. Bordered to the north by the River Po and to the south by the Apennine Mountains, the area is characterized by mountains in the southern and western portions and wide plains in the northern and eastern parts. The mountainous areas are occupied by the fold and trust belt of the Apennines, with the maximum altitude as 2165m (Figure 1a). This study focuses only on the mountain areas because they are extremely prone to landslides. The studied area has a typical Mediterranean climate: warm and dry summers and cool and wet winters.

The studied area suffers from a wide variety of landslide topologies, with the rainfall-induced landslides most common (Martelloni et al., 2012). Two kinds of rainfall are often associated with landslide events: the short but intense rainfall is likely to trigger shallow landslides, and deep-seated landslides are mainly influenced by the moderate but prolonged periods of rainfall (Ibsen and Casagli, 2004). Although landslides are not usually deadly, they are destructive. When landslides occur, the private and public properties, facilities and infrastructures are always exposed to the hazards, associated with the large cost of the regeneration and remedial works. Berti et al. (2012) mentioned that this kind of cost reached €130 million for 4 years from 2008 to 2012 in the Emilia-Romagna region. The abundance of landslides, as well as the availability of the required data, makes this region a good site to carry out this study.

### 2.2 Data Sources

The data used for the threshold definition includes daily rainfall, daily average temperature and daily soil moisture data. The daily rainfall is used to calculate the Antecedent Precipitation Index ($API_{v1}$) and the cumulated rainfall prior to landslide occurrences. The temperature and soil moisture information together with the daily rainfall are for the calculation of a modified Antecedent Precipitation Index ($API_{v2}$), which will be detailed in the following section. We collected all these data from ARPAE-ER (Regional Agency for the Prevention, Environment and Energy of Emilia Romagna), who maintains a hydro-meteorological network in the Emilia-Romagna region. This hydro-meteorological network could provide various data at different temporal scales, such as rainfall, air temperature, wind speed, relative humidity, etc. All these data can be obtained online (http://www.smr.arpa.emr.it/dext3r/). The rainfall and temperature data used in this study is from 50 weather stations, whose location is marked with the red triangle in Figure 1b. As for the soil moisture data, only the soil water content ($m^3/m^3$) at 10 cm soil depth of San Pietro Capofiume site is applied, due to its long-term records. The location of this site is marked with the yellow star in Figure 1b.

Our landslide inventory is provided by the Emilia-Romagna Geological Survey, who is responsible for maintaining a catalogue of historical landslides in the Emilia-Romagna region. The basic information recorded in the catalogue includes



the landslide occurrence location, date and the date accuracy level, which are complete for all events. More detailed information like landslide characteristics (length, width, type and material), triggering factors, damage and references are only available for part of the landslide events. These records rely on various sources, such as reports to local authorities, national and local press, technical documents, etc. Despite the rich source of information, the landslide inventory probably

represents a fraction of the actual landslide events, because some landslides with little damage or influences, especially those occurring in the remote area, are likely to be undetected or unreported. In this study, we only take advantage of the landslides with daily accuracy in terms of the occurrence date. Considering the completeness of all the required data, the study period is from 2006 to 2016, during which there are 168 landslides meeting the demand (Figure 1b). The 137 landslides during the period 2006 to 2014 (calibration period) are used for the threshold definition, and those of the period 2015 to 2016

(validation period) are for the threshold evaluation, with a total of 31.

## 3 Methods

### 3.1 Antecedent Precipitation Index (API)

Antecedent Precipitation Index (API) is employed in this study to characterize the wetness condition, which is derived from the preceding daily rainfall. It is noted that Antecedent Precipitation Index should be seen as a soil moisture index, allowing

us to estimate the relative wetness condition of the soil, which is sufficient for the aim of this study. One common definition of this index was proposed by Fedora (1987), written as:

$$API_t \ = \ k\,API_{t-\Delta t} + \ P_{\Delta t} \tag{1}$$

where $API_t$ is the API value at time t, $P_{\Delta t}$ is the cumulated precipitation during the period from $t - \Delta t$ to t (in this study $\Delta t = 1$ day), and k is the recession coefficient, less than 1, used to reflect the rate of drainage and evapotranspiration process.

Depending on the value of the recession coefficient, there are two versions of API. The first one ($API_{v1}$ hereafter) assumes the recession coefficient constant throughout the year, and the value of 0.84 is widely used in the previous researches, recommended by Crozier and Eyles (1980). The second version ($API_{v2}$ hereafter) allows the recession coefficient to vary according to the change of temperature, taking into account the effect of temperature on the evapotranspiration process. The variation of the recession coefficient is assumed to be linear in the work of Crow et al. (2005), which also applies in this

study:

$$k \ = \ 0.84 + \ \delta\,(20 - T_{ave}) \tag{2}$$

where $T_{ave}$ is the daily average temperature (℃) and $\delta$ is a sensitivity parameter (℃$^{-1}$). When $\delta = 0$, the recession coefficient is constant as 0.84. We used 20 ℃ as the basis as it is the most common temperature when the value of 0.84 is used. The sensitivity parameter $\delta$ is determined by comparing the $API_{v2}$ time series with the soil moisture data of San Pietro Capofiume site for the period from 2006 to 2014, where Pearson correlation coefficient is used as the evaluation criterion.



The optimized parameter is then validated using the data of the period 2015 to 2016. Given the study area has the similar variation pattern in terms of temperature, it is assumed the validated parameter at San Pietro Capofiume site could be extrapolated to the study area. The derived two versions of API are used to establish the threshold for landslide occurrence.

### 3.2 Thresholds for landslides

In order to explore the usage of the antecedent wetness information and the recent rainfall in landslide thresholds, we take advantage of the empirical threshold approach, which is carried out by analyzing the hydrological and meteorological conditions responsible for the occurrence of historical landslides. The meteorological condition as the final trigger is characterized by the recent 3-day cumulated rainfall prior to landslides. The hydrological condition (here is the wetness condition) is indexed with the API value. When calculating these variables responsible for the landslide, the data from the

nearest weather station are used. Based on the data of the period from 2006 to 2014, four types of thresholds are determined, as listed in Table 1.

1) Hybrid threshold

The hybrid threshold is established to explicitly include the antecedent wetness condition and the recent rainfall. The recent

rainfall is described by the most recent 3-day cumulated rainfall prior to landslide occurrences, while the antecedent wetness condition is indexed by the API value of the day prior to the recent 3 days. The hybrid threshold is determined by these two variables of the landslide events. Various combinations of these two variables are explored, where the API value and the recent 3-day cumulated rainfall is defined by their different percentiles. The percentile rank considered in this study includes 1, 2, 3, 4, 5, 6, 7, 8, 9, 10, 20 and 50. Taking the rainfall's 10th percentile (P10) as an example, it means that 10% landslides

have a 3-day cumulated rainfall less than P10. The higher the percentile rank, the stricter the threshold. We firstly used the API threshold as a cutoff, under which no landslide is expected. When it is exceeded, the 3-day cumulated rainfall threshold is then compared. The landslide occurrence is predicted only when these two threshold values are exceeded. Although there may be more complex approaches to define such hybrid threshold, like defining the threshold based on the functional relationship between these two variables (Mirus et al., 2018), the hybrid threshold proposed in this study could facilitate our

direct comparison of the threshold only based on the 3-day cumulated rainfall. As a result, we could investigate the direct impact of incorporating the antecedent wetness condition to the threshold definition.

2) 3-day rainfall threshold

The 3-day rainfall threshold is only based on the recent 3-day cumulated rainfall prior to landslide occurrences. It disregards

the antecedent wetness condition and only focuses on the recent rainfall prior to landslides. Various threshold values are explored, which are defined by its different percentiles, the same as the 3-day rainfall component in the hybrid threshold.

3) API threshold





API threshold is determined using the API value prior to the landslide occurrence. This variable is considered to include the antecedent wetness information and the recent rainfall information. In other words, the API threshold implicitly includes the effect of the antecedent wetness and the recent rainfall. The different percentiles of the API value prior to landslide occurrences are calculated as the threshold values.

4) Updated API threshold

The updated API threshold is based on the API threshold and updated with an added rule. In order to more explicitly consider the effect of the recent rainfall, the rule added is whether there is rainfall in the recent 3 days. From Equation (1), it is clear that as the recession coefficient is less than 1, if there is no rainfall, the API value will decrease. Therefore, if the API
value of the recent 3 days shows a decrease trend, even the API threshold is exceeded, the landslide occurrence will not be predicted. In contrast, if there is an increase trend of API value during the recent 3 days and the API threshold is exceeded, the landslide occurrence is predicted. The updated API threshold could take into account the effect of the recent rainfall.

With these four types of thresholds, three scenarios are designed, as listed in Table 2. Scenario 1 compares the prediction
performance of the hybrid threshold with that of the 3-day rainfall threshold, which is designed to explore the effect of incorporating the antecedent wetness information to the definition of landslide thresholds. Scenario 2 and Scenario 3 are designed to answer whether it is necessary to explicitly consider the antecedent wetness condition and the recent rainfall in the threshold definition. The prediction performance of the hybrid threshold is compared with that of the API threshold in Scenario 2. The comparison between the API threshold and the updated API threshold is carried out in Scenario 3.

**3.3 Threshold evaluation**

The prediction performance of different thresholds is evaluated using the data of the period 2015 to 2016. For this procedure, we only select the weather stations whose vicinities have landslide events. The contingency matrix and Receiver Operating Characteristic (ROC) curves are applied for the purpose, which are the most common tools used for the threshold evaluation (Gariano et al., 2015;Mirus et al., 2018;Staley et al., 2013).

The contingency matrix consists of four components: Ture Positive (TP), False Negative (FN), False Positive (FP) and True Negative (TN), which are the four possible outcomes of the thresholds' prediction results. These prediction results are based on a fixed daily interval from 0:00 A.M.-11:59 P.M. local time. TP events are when the threshold is exceeded and one or more landslides occur. FN events are when the threshold is not exceeded, but there are one or more landslides; FP events are
when the threshold is exceeded, but on landslides occur. TN events are when the threshold is not exceeded and there are no landslides.





Receiver Operating Characteristic (ROC) curve is plotted with Hit Rate against False Alarm Rate. Hit Rate (HR) is also known as the true positive rate, which is used to measure the proportion of landslides that are correctly predicted. It can be calculated as:

$$HR = \frac{TP}{TP + FN} \qquad (3)$$

False Alarm Rate (FAR) is also known as the false positive rate, which is used to measure the proportion of false alarms over the events when no landslide occurs. It can be calculated as:

$$FAR = \frac{FP}{FP + TN} \qquad (4)$$

The variation range of HR and FAR is both from 0 to 1. The optimal value of HR and FAR is 1 and 0, respectively. Therefore, the optimal prediction performance is achieved when HR equals 1 and FAR equals 0 (perfect point). In reality, it is difficult for a threshold to reach the perfect point, as a result, the Euclidean distance (d) to the perfect point is used as a criterion to evaluate the prediction performance (Gariano et al., 2015). The smaller the distance, the better the performance. Sometimes the HR value needs to be restricted as 1 owing to the danger of missed alarms. In this case, the smaller the FAR value, the better the prediction performance.

For each threshold approach, we explored various values or combinations. In order to evaluate the predictive capability of one certain threshold approach, the area under the ROC curve (AUC) of the threshold approach is calculated. The larger the area, the better the predictive capability.

## 4 Results

### 4.1 Thresholds for landslides

Before analyzing the hydrological and meteorological conditions responsible for the occurrence of historical landslides, we firstly test the reliability of API in indexing the wetness condition. The sensitivity parameter $\delta$ of $API_{v2}$ is calibrated as 0.006 $°C^{-1}$. In order to validate the parameter, its performance is evaluated using the data of the independent period 2015 to 2016. Figure 2 shows the scatter plot of API against the soil moisture data at San Pietro Capofiume site, with Figure 2a for $API_{v1}$ and Figure 2b for $API_{v2}$. The Pearson correlation coefficient ($r_p$) is 0.71 for the $API_{v2}$. Although it can't be considered significant, it shows a great improvement compared with $API_{v1}$, whose Pearson correlation coefficient ($r_p$) is 0.51. From the data distribution in Figure 2b, it is seen that the poor linear relationship is mostly attributed to the high values of $API_{v2}$. The soil water content is limited by the maximum water capacity of the soil layer; however, there is no restriction for the $API_{v2}$ value. Therefore, if the points with high $API_{v2}$ values are restricted by a maximum value, the linear relationship between the $API_{v2}$ value and soil water content will become more significant. As the API value is employed to index the relative soil





wetness state, we also calculated the Spearman's rank correlation coefficient, which could measure the statistical dependence between the rankings of two variables. It is found that the Spearman's rank correlation coefficient is high (0.82), indicating that there is a similar rank between the $API_{v2}$ value and soil water content. Therefore, the parameter of $API_{v2}$ could be regarded as acceptable, and we use it to calculate $API_{v2}$ value of all landslides. As for the $API_{v1}$, both Pearson and Spearman

correlation coefficient are low, implying the poor relationship between $API_{v1}$ and soil water content. Despite this, we also calculated $API_{v1}$ for the comparison purpose.

The distribution of landslides' variables (as listed in Table 1) is shown in Figure 3 and Figure 4. Figure 3a is for the API value of the day prior to the recent 3 days, Figure 3b is for the recent 3-day cumulated rainfall prior to landslide occurrences,

and Figure 4 is for the API value prior to landslide occurrences. From Figure 3a, the API value of the day prior to the recent 3 days is higher for $API_{v2}$ than $API_{v1}$. This is due to the variation of the recession coefficient in $API_{v2}$. The Emilia-Romagna region is characterized by the Mediterranean climate, with warm and dry summers and cool and wet winters. For rainfall-induced landslides, they mostly occur in the wet season, during which the temperature is low. According to Equation (2), the recession coefficient in the season is likely to be higher than 0.84, and the lower loss rate of the preceding rainfall leads to a

higher API value. The similar result can be found in Figure 4. It is found all these three variables have a wide variation range. Taking the 3-day cumulated rainfall as an example, the amount of 0.4 mm is likely to trigger landslides, while the amount of 231.2 mm is also responsible for the landslide initiation. This indicates in several cases with small rainfall amount, the occurrence of landslides is not just attributed to the recent rainfall prior to landslides, where the antecedent wetness condition plays a key role. The variables' wide variation range implies that the conditions responsible for landslides

varies a lot, which is also the reason why different threshold values are explored for each threshold approach.

Different percentiles of landslides' variables (represented by triangles in Figure 3 and Figure 4) are calculated for the threshold definition, whose value is listed in Table 3. The percentile rank considered in this study includes 1, 2, 3, 4, 5, 6, 7, 8, 9, 10, 20 and 50. The higher the percentile rank, the greater the threshold value. With these percentiles, four types of

thresholds are determined following the procedure described in Section 3.2.

## 4.2 Threshold comparison

To compare the thresholds in the three scenarios listed in Table 2, their prediction performance is evaluated by calculating contingencies and preforming the ROC analysis. In the ROC plot, the line represents the performance of one certain threshold approach, and the area under the ROC curve (AUC) could measure its predictive capability. The points on each

line from right to left represent different threshold levels, which are defined at the percentile rank of 1, 2, 3, 4, 5, 6, 7, 8, 9, 10, 20 and 50, respectively. For the hybrid threshold that has two components, the points on the line represent the variation of the 3-day rainfall's threshold level, where the API's threshold is defined with its 10th percentile. We also explored 12



different percentiles of the API, the optimal performance is achieved at the 10th percentile rank with AUC as 0.92 for $API_{v1}$ and 0.96 for $API_{v2}$, so only this case is presented and analyzed.

For this procedure, the data of the period from 2015 to 2016 are used. There are 12 weather stations whose vicinities have landslide events. The landslides occurring in the same day and belonging to the same weather station are regarded as one landslide event, which is the reason why the 31 landslides are only counted as 22 landslide events in the validation procedure. As we performed the landslide prediction at a daily interval, there should be 8772 contingencies in total for the 12 weather stations. However, due to the missing data of a few days, only 8745 contingencies are obtained for each threshold.

a) Scenario 1

The prediction results of the hybrid threshold and the 3-day rainfall threshold are compared in Figure 5, with Figure 5a for $API_{v1}$, and Figure 5b for $API_{v2}$. It is clear that with the increment of the threshold, the false alarm rate is reduced sometimes at the expense of decreasing the hit rate. For the hybrid threshold based on $API_{v1}$ (Figure 5a), its AUC value is a little smaller than that of the 3-day rainfall threshold, which is unexpected given the important role of the antecedent soil moisture condition in the initiation of landslides. It is clear that this is mainly due to the missed alarms caused by the lower hybrid threshold levels. However, as for the false alarm rate, the hybrid threshold presents a great improvement compared with the 3-day rainfall threshold. The right plot in Figure 5a shows the proportion of the reduced false alarms which is attributed to the added antecedent wetness information. It is clear that the lower the threshold level in terms of the 3-day cumulated rainfall is, the higher the proportion of reduced false alarms is. This indicates the False Positive (FP) contingencies predicted by the lower 3-day rainfall threshold have a higher proportion of the dry antecedent wetness condition. The FP contingencies with dry antecedent wetness condition are excluded by adding the API information, and thus has a higher proportion of reduced false alarms. In contrast, the False Positive (FP) contingencies predicted by the higher 3-day rainfall threshold level have a lower proportion of the dry antecedent soil wetness condition. Therefore, it is implied that considering the antecedent wetness condition is more crucial when using the lower 3-day rainfall threshold. The above results also apply to the case of $API_{v2}$ in Figure 5b except that the AUC value of the hybrid threshold is a littler smaller than that of the 3-day rainfall threshold. From Figure 5b, it is expected that the hybrid threshold based on $API_{v2}$ has a higher AUC value than the 3-day rainfall threshold. Based on the opposite result from the case of $API_{v1}$, it is implied that $API_{v2}$ has better representativeness of the soil wetness condition than $API_{v1}$, which is in line with the results in Figure 2. By comparing the hybrid threshold based on $API_{v2}$ with that based on $API_{v1}$, it is found the hybrid threshold based on $API_{v2}$ not only increases the hit rate, but also improves the performance of

b) Scenario 2

Figure 6 shows the prediction results of the hybrid threshold and the API threshold, with Figure 6a for the thresholds based on $API_{v1}$, and Figure 6b for the thresholds based $API_{v2}$. By analyzing the AUC value, it is found that for both $API_{v1}$ and





$API_{v2}$, the AUC value of the hybrid threshold is greater than that of the API threshold, and the improvement is more distinct for $API_{v1}$ than $API_{v1}$. From Figure 6a, although the hybrid threshold is more capable of reducing the false alarm rate, its hit rate of the lower threshold level is smaller, which influences the AUC value. As for the thresholds based on $API_{v2}$, the hybrid threshold not only reduces the false alarm rate, its performance of hit rate is also superior to API threshold.

c) Scenario 3

The comparison results of the API threshold and the updated API threshold are shown in Figure 7. Figure 7a is for the thresholds based on $API_{v1}$, and Figure 7b is for the thresholds based on $API_{v2}$. From Figure 7a, these two threshold approaches have the same AUC value, while for $API_{v2}$ in Figure 7b, the updated API threshold has a larger AUC value than the API threshold. It is found that for both $API_{v1}$ and $API_{v2}$, the updated API threshold has a superior performance in reducing the false alarm rate, which is clear in the right bar plot. With the increase of the threshold level, the proportion of reduced false alarms which are caused by the updated API threshold decreases. This indicates that among the False Positive (FP) contingencies predicted by the lower API threshold, there is a higher proportion of the cases without rainfall during the recent 3 days. In contrast, for the False Positive (FP) contingencies predicted by the higher API threshold, there is a lower proportion of the cases without rainfall during the recent 3 days. Therefore, highlighting the role of the recent 3-day rainfall is more important when the lower API threshold level is used. By comparing Figure 7b with Figure 7a, it is clear that the updated API threshold's ability to reduce false alarms is superior for the $API_{v2}$ version to $API_{v1}$ version.

**4.3 The optimal threshold**

To determine the optimal threshold level for each threshold approach, the Euclidean distance is used as the criterion to measure the balance between the correct predictions and incorrect predictions. The optimal prediction results determined by the smallest Euclidean distance are listed in Table 4. Among the seven optimal thresholds, the hybrid threshold based on $API_{v2}$ has the smallest distance to the perfect point, with the hit rate as 0.95 and false alarm rate as 0.11. The updated API threshold based on $API_{v2}$ could also provide a better prediction result, where the hit rate is 0.91 and the false alarm rate is 0.10. It is interesting to find that these two threshold definition approaches could explicitly consider the antecedent wetness condition and the recent rainfall. The superiority of these two threshold approaches is mainly reflected in reducing the false alarm rate, though the improvement in terms of the hit rate value is more distinct. This is because the landslide events used for the validation procedure are very limited, even a small variation in the True Positive (TP) contingency will lead to an obvious variation in the value of hit rate. Taking the two versions of the hybrid threshold as an example, although the hit rate increases from 0.91 to 0.95, this is caused only by the difference of one TP contingency. However, for the false alarm rate, the decease from 0.15 to 0.11 needs a difference of 266 False Positive (FP) contingencies. It is also found the optimal thresholds determined using the $API_{v2}$ data could provide better performance than those based on $API_{v1}$ data.



In practice, in order to avoid the risk of missed alarms, the hit rate is encouraged to be 1. In this case, the optimal prediction results are achieved with the smallest false alarm rate. Table 5 lists the optimal results determined in this way. The hybrid threshold based on $API_{v1}$ fails to have the optimal result when the hit rate is restricted to 1, since all its cases have a hit rate less than 1. Among the rest five threshold versions, the hybrid threshold and the updated API threshold determined using $API_{v2}$ also provide the best results. Their false alarm rate is improved obviously compared with other threshold approaches, with 0.16 for the updated API threshold and 0.17 for the hybrid threshold. It is also found that using $API_{v2}$ data in the definition of threshold could benefit its prediction performance, compared with $API_{v1}$.

## 5 Discussion

The results of the designed scenarios provide useful information for the landslide threshold definition by combining the antecedent wetness information and the recent rainfall. For the hybrid threshold and the 3-day rainfall threshold in Scenario 1, the only difference between them is whether to include the antecedent wetness information. Therefore, comparing the prediction results of these two thresholds could help investigate the direct impact of incorporation the antecedent wetness information to the threshold definition. The results show that the hybrid threshold is capable of improving the false alarm rate compared with the 3-day rainfall threshold, indicating that considering both antecedent wetness information and the recent rainfall in the threshold definition is more in line with the physical process. This implies the crucial role of the antecedent wetness condition in the occurrence of landslides. Based on the acknowledgement of the key role of the antecedent wetness condition and the recent rainfall, there is a question of how to take advantage of these two types of information in the threshold definition. To answer this question, Scenario 2 and Scenario 3 are designed. The hybrid threshold in Scenario 2 and the updated API threshold in Scenario 3 could be regarded as the case which could explicitly consider the antecedent wetness information and the recent rainfall. Although the API threshold in Scenario 2 and Scenario 3 includes these two types of information, the information is integrated into one variable. In other words, for the API threshold, the antecedent wetness condition and the recent rainfall is not considered explicitly. Therefore, by designing Scenario 2 and Scenario 3, we could answer whether it is necessary to explicitly consider the antecedent wetness condition and the recent rainfall in the threshold definition. Results of Scenario 2 show that the hybrid threshold could provide a better prediction performance in terms of increasing hit rate and reducing false alarm rate. In Scenario 3, by explicitly considering the recent rainfall, the updated API threshold presents a distinct improvement in reducing false alarms compared with the API threshold. Based on these results, it is concluded that explicitly considering the antecedent wetness condition and the recent rainfall in the threshold definition could improve the threshold's prediction performance. In addition, the prediction results of the optimal thresholds for each threshold approach also show that the optimal hybrid threshold and the updated API threshold have the best prediction performance, with higher hit rate and lower false alarm rate.





When defining the threshold, two versions of API are used to index the soil wetness state. The recession coefficient remains constant for $API_{v1}$, while the recession coefficient is allowed to vary according to the change of the temperature for $API_{v2}$. By comparing $API_{v1}$ and $API_{v2}$ with soil water content at San Pietro Capofiume site, respectively, it is found that $API_{v2}$ is more correlated with the soil water content. The $API_{v2}$'s better representativeness of the soil moisture is also reflected in the threshold performance, where the thresholds based on $API_{v2}$ present better prediction results than those based on $API_{v1}$. This indicates when including the soil moisture information in the threshold definition, the reliability of the soil moisture measurement is also a key factor affecting the threshold predictive capability.

Although the above results could provide useful information for the landslide threshold definition, it is noted the method we employed in this study is based on the statistical approach. Therefore, the proposed results probably be influenced by the data used for the threshold evaluation, which is also highlighted in the work of Gariano et al. (2015). They stated that the lack of landslide information has a great impact on the contingencies and the skill scores used to evaluate the threshold forecasting performance. In our study, the considered landslides are likely to be incomplete, which will cause the uncertainties to the contingencies and the ROC analysis. However, given the large proportion of the days without landslides (the sum of False Positive (FP) contingencies and True Negative (TN) contingencies), according to Equation (4) the variation in the landslide events has little impact on the false alarm rate. From the results of the thresholds, the improvement caused by adding antecedent soil wetness information (or explicitly including two types of information) mainly reflects in reducing false alarms. As a result, it is regarded the proposed results are robust. Despite this, explorations with more complete data are encouraged to test the proposed results. To better understand the role of the antecedent wetness condition and the recent rainfall in the occurrence of rainfall-induced landslides, a physical-based approach is expected. The physical process understanding could help construct the threshold which is more in line with the practice and improve the predictive capability.

In addition to the results on the usage of the antecedent wetness condition and the recent rainfall, we also find the updated API threshold has better prediction performance, which provides a new perspective for the threshold definition for rainfall-induced landslides. When defining the updated API threshold, only one variable (API) is needed, which could avoid the construction of the function relationship between two variables of the rainfall threshold, like the power law of the rainfall intensity-duration threshold. Besides, the updated API threshold could take into consideration both the antecedent wetness condition and the recent rainfall, which proves to be beneficial for the threshold's predictive capability. Though we employed API to index the soil moisture condition, this threshold definition approach could apply to other measures of the soil moisture.

There are other points worth noting. First, when separate the antecedent wetness condition from the recent rainfall, 3 days are selected as the boundary. Although there may be many other selections for this separation, the initial exploration we

present here is intended as a proof-of-concept. Therefore, we start by using 3 days as the separation to explore the usage of the antecedent wetness condition and the recent rainfall in the threshold definition. Besides, 3 days are widely used to separate the antecedent condition from the recent condition in the previous studies (Chleborad et al., 2008;Scheevel et al., 2017;Mirus et al., 2018). Second, Antecedent Precipitation Index (API) is used as a proxy of soil moisture in this study,

owing to its simple formulation and less data input. Although we try to improve the API's representativeness of the soil moisture by allowing the recession coefficient to vary, it can only be regarded as an indicator of the soil moisture. Therefore, to make the proposed results more reliable, explorations based on more accurate measures of the soil moisture are encouraged.

## 6 Conclusion

The empirical rainfall threshold is employed as the tool to explore the usage of the antecedent wetness condition and the recent rainfall in the landslide threshold. Four types of thresholds are proposed, by including different variables that are responsible for landslide occurrences. These thresholds could represent different cases, like whether to include the antecedent wetness condition or whether to consider the recent rainfall explicitly. The predictive capability of these thresholds is compared crossly based on three scenarios. The following conclusions can be drawn from the comparison

results:

1. The antecedent wetness condition plays a crucial role in the occurrence of rainfall-induced landslides. As a result, incorporating the antecedent wetness information in the threshold definition could improve the threshold's prediction performance in terms of reducing false alarms.

2. It is beneficial for the threshold's predictive capability to explicitly include the antecedent wetness information and the

recent rainfall in the definition of thresholds for rainfall-induced landslides.

3. When including soil moisture information in the landslide threshold, the reliability of the soil moisture measurement is a key factor affecting the threshold's predictive capability.

We expect these conclusions could increase the attention to the role of the antecedent wetness condition and the recent rainfall when defining thresholds for rainfall-induced landslides.

**Acknowledgement**

The authors acknowledge Dr. Matteo Berti for providing landslides data and Arpae Emilia-Romagna organization for providing the meteorological data. The first author would like to thank the China Scholarship Council for funding her study at the University of Bristol. This work is supported by the National Natural Science Foundation of China (41871299), in part by Resilient Economy and Society by Integrated SysTems modelling (RESIST) (Newton Fund via Natural Environment





Research Council (NERC) and Economic and Social Research Council (ESRC) (NE/N012143/1)), and in part by the
Fundamental Research Funds for the Central Universities of China (2016B42014).

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





## Tables

**Table 1. Four types of the threshold definition approach**

| No. | Threshold Type | Variables |
|-----|----------------|-----------|
| 1 | Hybrid threshold | the recent 3-day cumulated rainfall prior to landslide occurrences |
| | | the API value of the day prior to the recent 3 days |
| 2 | 3-day rainfall threshold | the recent 3-day cumulated rainfall prior to landslide occurrences |
| 3 | API threshold | the API value prior to landslide occurrences |
| 4 | Updated API threshold | the API value prior to landslide occurrences |
| | | an increase trend of API value during the recent 3 days |

**Table 2. The scenarios for the threshold comparison**

| No. | Scenario | Aim |
|-----|----------|-----|
| 1 | Hybrid threshold | What is the effect of incorporating the antecedent |
| | 3-day rainfall threshold | wetness information to the threshold definition? |
| 2 | Hybrid threshold | Is it necessary to explicitly consider the antecedent |
| | API threshold | wetness condition and the recent rainfall in the threshold |
| 3 | API threshold | definition? |
| | Updated API threshold | |

**Table 3. The percentiles of landslides' three variables**

| Label | API prior to the 3 days (mm) | | 3-day cumulated rainfall (mm) | API prior to landslides (mm) | |
|-------|------|------|------|------|------|
| | $API_{v1}$ | $API_{v2}$ | | $API_{v1}$ | $API_{v2}$ |
| P1 | 3.11 | 10.64 | 0.58 | 7.75 | 15.03 |
| P2 | 3.56 | 18.16 | 1.03 | 10.70 | 24.22 |
| P3 | 4.61 | 18.69 | 1.72 | 12.97 | 43.83 |
| P4 | 6.01 | 19.61 | 2.45 | 14.66 | 46.32 |
| P5 | 6.57 | 22.81 | 2.60 | 15.90 | 49.70 |
| P6 | 6.90 | 24.58 | 2.74 | 16.03 | 50.65 |
| P7 | 7.55 | 27.96 | 3.63 | 17.07 | 51.71 |
| P8 | 8.72 | 30.00 | 4.01 | 18.51 | 52.79 |
| P9 | 9.90 | 32.06 | 4.58 | 19.27 | 54.84 |
| P10 | 11.01 | 33.87 | 5.14 | 19.58 | 57.11 |
| P20 | 15.66 | 55.00 | 12.60 | 28.16 | 72.48 |
| P50 | 36.72 | 91.13 | 36.00 | 59.52 | 117.70 |





**Table 4. The prediction results of the optimal thresholds determined by the smallest Euclidean distance**

| Optimal Threshold | | Percentile | TP | FN | FP | TN | HR | FAR | d |
|---|---|---|---|---|---|---|---|---|---|
| Hybrid threshold | $API_{v1}$ | P20 | 20 | 2 | 1266 | 7457 | 0.91 | 0.15 | 0.17 |
| | $API_{v2}$ | P20 | 21 | 1 | 1000 | 7723 | 0.95 | 0.11 | 0.12 |
| 3-day rainfall threshold | | P20 | 21 | 1 | 1815 | 6908 | 0.95 | 0.21 | 0.21 |
| API threshold | $API_{v1}$ | P20 | 20 | 2 | 1910 | 6813 | 0.91 | 0.22 | 0.24 |
| | $API_{v2}$ | P20 | 20 | 2 | 1482 | 7241 | 0.91 | 0.17 | 0.19 |
| Updated API threshold | $API_{v1}$ | P20 | 19 | 3 | 1364 | 7359 | 0.86 | 0.16 | 0.21 |
| | $API_{v2}$ | P20 | 20 | 2 | 860 | 7863 | 0.91 | 0.10 | 0.13 |

**Table 5. The prediction results of the optimal thresholds determined by restricting the hit rate to 1**

| Optimal Threshold | | Percentile | TP | FN | FP | TN | HR | FAR | d |
|---|---|---|---|---|---|---|---|---|---|
| Hybrid threshold | $API_{v1}$ | NULL | - | - | - | - | - | - | - |
| | $API_{v2}$ | P10 | 22 | 0 | 1465 | 7258 | 1 | 0.17 | 0.17 |
| 3-day rainfall threshold | | P10 | 22 | 0 | 2939 | 5784 | 1 | 0.34 | 0.34 |
| API threshold | $API_{v1}$ | P6 | 22 | 0 | 3582 | 5141 | 1 | 0.41 | 0.41 |
| | $API_{v2}$ | P4 | 22 | 0 | 2530 | 6193 | 1 | 0.29 | 0.29 |
| Updated API threshold | $API_{v1}$ | NULL | - | - | - | - | - | - | - |
| | $API_{v2}$ | P4 | 22 | 0 | 1389 | 7334 | 1 | 0.16 | 0.16 |





**Figures**

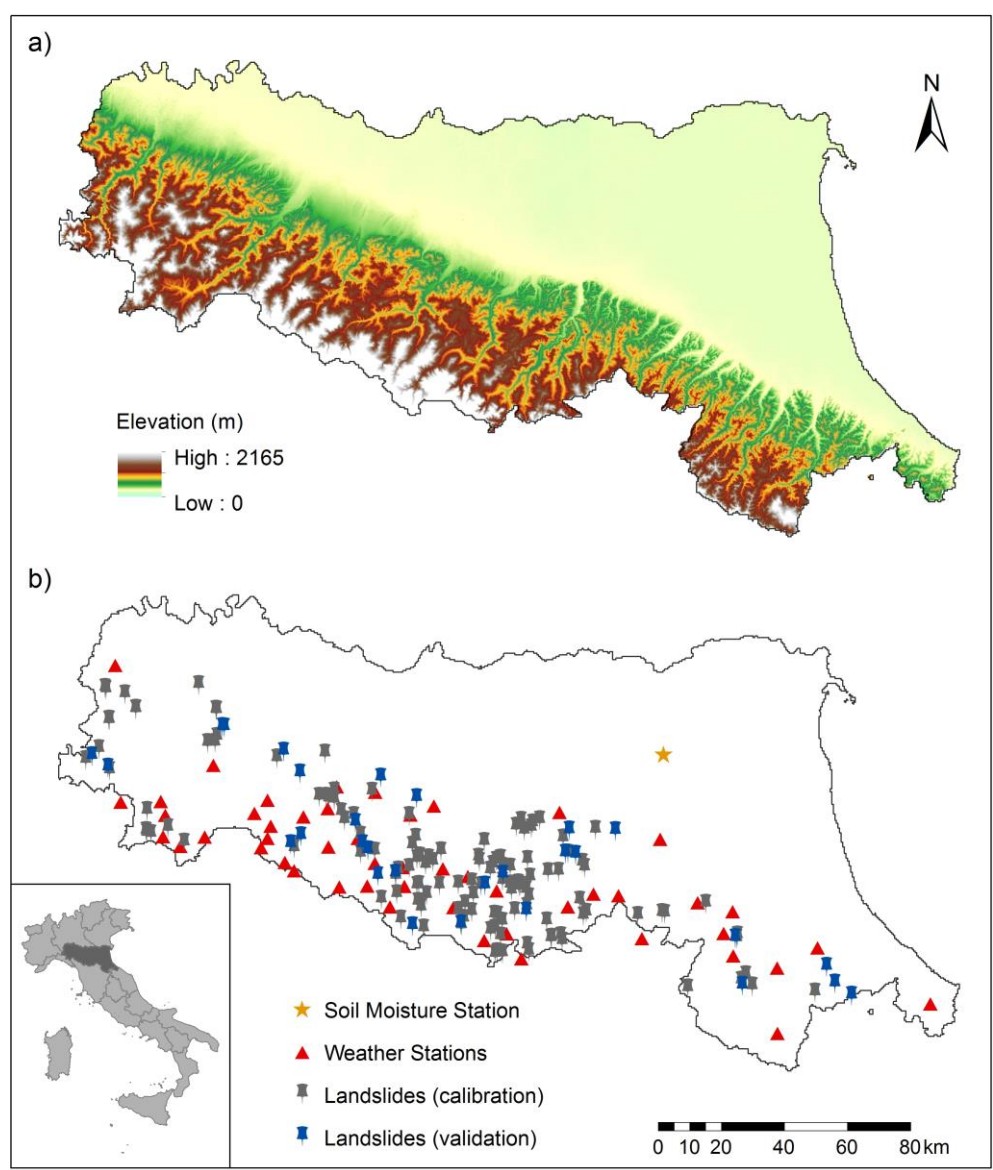

**Figure 1. a) Location of Emilia-Romagna region with its DEM map and b) distribution of studied landslides and in-situ measurement stations**



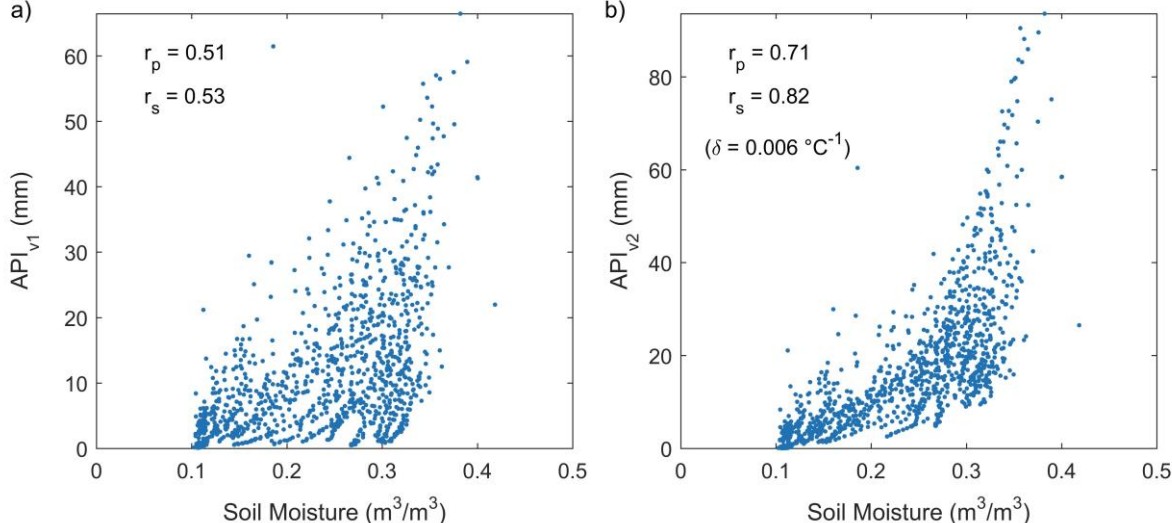

**Figure 2. The scatter plot of API against the soil moisture at San Pietro Capofiume site, a) for API$_{v1}$, b) for API$_{v2}$**



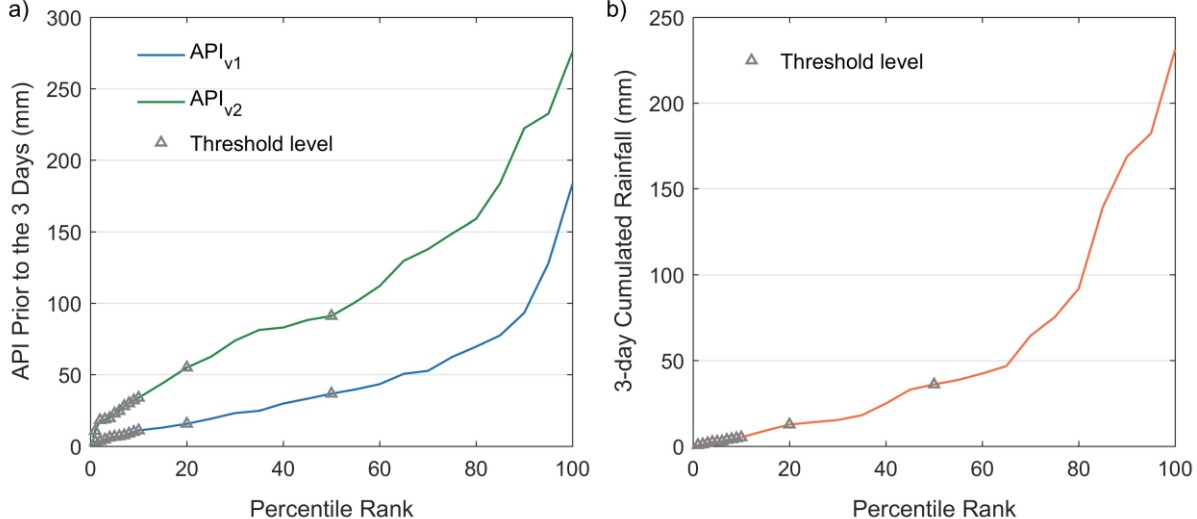

**Figure 3. The distribution of two variables of landslides as well as their different percentiles, a) for the API value of the day prior to the recent 3 days, b) for the recent 3-day cumulated rainfall prior to landslide occurrences**





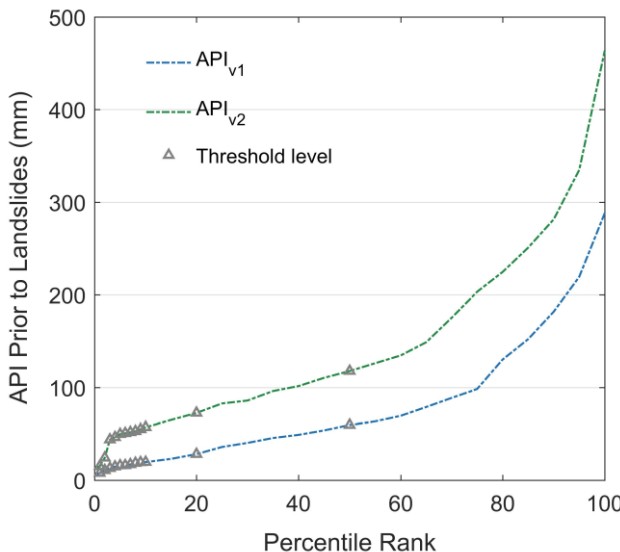

**Figure 4. The distribution of the API value prior to landslide occurrences as well as its different percentiles**





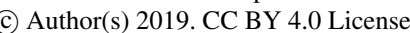

**Figure 5. The prediction results of the hybrid threshold and 3-day rainfall threshold, a) for the hybrid threshold based on APIv1, b) for the hybrid threshold based on APIv2**





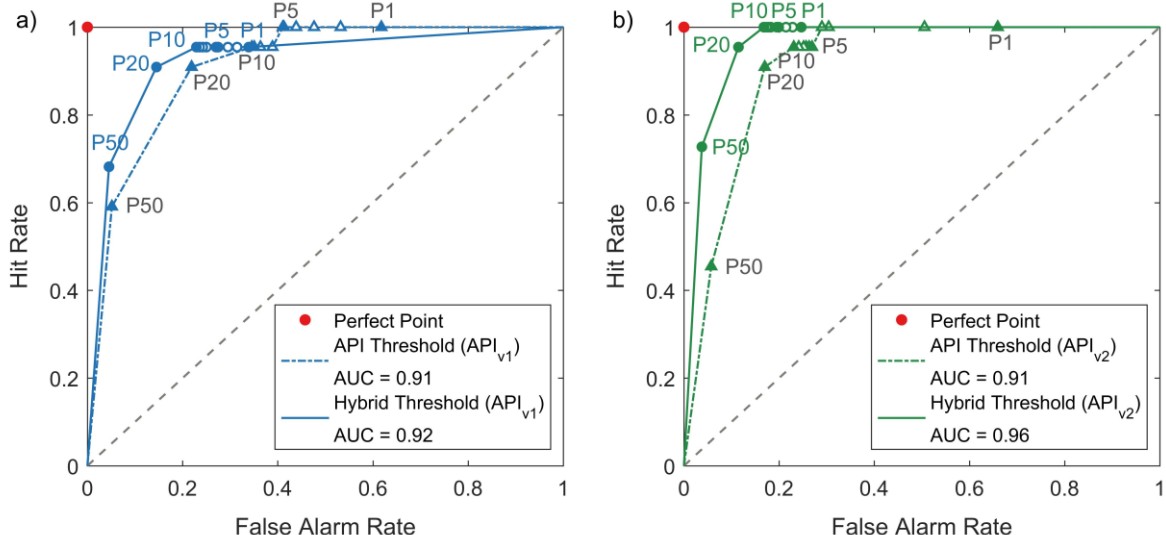

**Figure 6. The prediction results of the hybrid threshold and API threshold, a) for the thresholds based on API$_{v1}$, b) for the thresholds based on API$_{v2}$**

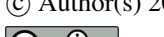



**Figure 7. The prediction results of the API threshold and the updated API threshold, a) for the thresholds based on API_v1, b) for the thresholds based on API_v2**

