# Peer review of "Antecedent wetness and rainfall information in landslide threshold definition"

_Hydrology and Earth System Sciences, 2019_

## Referee Comment (RC1) · Dr. Max Oke Kluger (Referee) · 18 Apr 2019

**GENERAL COMMENTS**

In the paper under consideration, the authors present rainfall characteristics, soil moisture data, and landslide inventories collected in North Italy from 2006 to 2016. The commonly used thresholds based on antecedent precipitation index (API-threshold) and 3-day cumulated rainfall (3-day-threshold) are compared with two new thresholds. The hybrid threshold combines information from the established API- and 3-day-thresholds, whereas the updated API threshold considers positive or negative trends in the API preceding the landslide event.

In general, I see some merit in the paper as it addresses important aspects of rainfall-

triggered landslides and how to improve risk predictions in regions prone to strong rainfall events. However, I consider the present manuscript needs substantial improvements in order to be publishable. During my revision I focused on the following issues: (1) Is the study sufficiently motivated by literature? (2) Are the methods sufficiently described to allow others replicating their work? (3) Are the results presented in a comprehensive way? And (4) are the conclusions supported by the results of the paper?

SPECIFIC COMMENTS

(1) Is the study motivated by the literature?

[p.1 – p.3 | Introduction] The authors introduce the rainfall triggering of landslides and give examples. They introduce existing rainfall thresholds in detail and support their motivation for the paper with literature. In my opinion, there is some potential for shortening, though.

Conclusion 1: the paper is well motivated by literature and furthermore fits well within the framework of the journal.

(2) Are the methods sufficiently described to allow others replicating their work?

[p.5 | L13-L19] The reference cited for equation 1 is a M.Sc. thesis and not an adequate reference, as it doesn't have any proper ISI citations. I wonder, if there is a better paper to cite here as the API is a well-known equation. The authors did not define how API itself is derived from rainfall data. This would be helpful for readers not working with API on a daily basis.

[p.6 | L14-L26] I had a hard time understanding this paragraph and am not sure I fully understood it in the end. The authors write 'The hybrid threshold is established to explicitly include the antecedent wetness condition and the recent rainfall. [...] antecedent wetness condition is indexed by the API value of the day prior to the recent 3 days. [...] Various combinations of these two variables are explored, where the API

value and the recent 3-day cumulated rainfall is defined by their different percentiles.' What do the authors mean exactly with 'explicit'? Why do the authors use the API of the day prior to the recent 3 days? What do the authors mean with various combinations of both variables in respect to the percentiles? The authors proceed with 'The percentile rank considered in this study includes 1, 2, 3, 4, 5, 6, 7, 8, 9, 10, 20 and 50.' However, Figs. 3 and 4 display percentile ranks up to 100%. From the text it is unclear to me how the authors calculated the API curves (e.g. green and blue curve in Fig. 3a). The authors proceed with 'We firstly used the API threshold as a cutoff, under which no landslide is expected. When it is exceeded, the 3-day cumulated rainfall threshold is then compared. The landslide occurrence is predicted only when these two threshold values are exceeded.' What do the authors mean with 'cutoff' and 'landslide occurrence is predicted'? I advise the authors to improve this paragraph by adding more details and a schematic figure explaining how the threshold(s) work. I think a basic figure of rainfall characteristics and occurrence of landslides would also be beneficial to better understand the threshold.

[p.6-p.7 | L28-L4] Maybe move these thresholds to the beginning as they are already established in the literature.

[p.7 | L6-L12] Similar to the paragraph explaining the hybrid threshold, this paragraph lacks in detail and clarity. It is not clear how this 'added rule' works in practice. What do the authors mean with 'more explicitly consider the effect of the recent rainfall'? I wonder whether the hybrid or the updated API threshold is more explicit?

[p.7 | L14-L19] This paragraph describes how the thresholds are compared between each other. I wonder why the authors did not compare each combination but only three? At least, it would be helpful, if the authors would point out the reason for their choice.

Conclusion 2: The methods could be much better explained by the authors. One may not be able to fully understand all steps required for data evaluation as well as the

threshold definitions.

(3) Are the results presented in a comprehensive way?

[p.9 | L8-L20] The authors write 'The distribution of landslides' variables (as listed in Table 1) is shown in Figure 3 and Figure 4.' What do the authors mean with 'landslide variables'? The table only lists the types of rainfall thresholds used in the study. 'Variables' is ambiguous. The Figs. 3 and 4 show API and 3-day rainfall vs percentile rank and no landslide variables. The authors proceed with 'Figure 3a is for the API value of the day prior to the recent 3 days, Figure 3b is for the recent 3-day cumulated rainfall prior to landslide occurrences, and Figure 4 is for the API value prior to landslide occurrences.' As mentioned before, the plots confuse me because it was not well explained how the authors calculated API or 3-day rainfall for percentiles between 1 and 100% (lines in Figs. 3 and 4). For me it was hard to connect Figs. 3 and 4 to the four thresholds listed in Tab. 1. I therefore suggest to have four plots showing each of the four thresholds independently. The authors write 'For rainfall - induced landslides, they mostly occur in the wet season, during which the temperature is low.' One can hardly judge about this statement because the authors did not provide time scales of rainfall and landslide occurrences. Later the authors write 'Taking the 3-day cumulated rainfall as an example, the amount of 0.4 mm is likely to trigger landslides, while the amount of 231.2 mm is also responsible for the landslide initiation.' I cannot find these numbers in the plot. Why is 0.4 mm rainfall more 'likely to trigger landslides' than 232.2 mm, which is mentioned as only 'responsible for the initiation of landslides'? Please revise the argumentation.

[p.9 | L21-L25] This is redundant to methods and I advise to delete it.

[p.9 – p.11 | Threshold comparison] I had a hard time understanding the Figs. 5 and 6 and the text corresponding to these figures. In my opinion, the text is too technical and requires more details when explaining the principle of the plots. For example, I did not understand why plotting the FAR in Fig. 5b, whereas in Fig. 5a the curves lie more

closely to the Hit Rate axis. In addition, what do the authors mean with 'threshold level of 3-day cumulated rainfall' in Fig. 5b? The bars reflect the percentiles not only for 3-day rainfall threshold but also for the hybrid threshold.

Conclusion 3: The presentation of results could be improved.

(4) Are the conclusions supported by the results of the paper?

[p.12-.14 | Discussion] In my opinion, the discussion needs to be improved. The authors did not critically reflect their results and compared them with other results from the literature. Generally, the discussion section requires more references.

[p.14 | L10-L24] In my opinion conclusions 1 and 2 are more or less identical. The only real conclusion the authors make here is '[. . .] could improve the threshold's prediction performance in terms of reducing false alarms.' Just stating that the antecedent wetness is important is not enough because this has been shown by numerous researchers in the past. The third conclusion is fine.

Conclusion 4: The conclusions are supported by the results, but need to be more specific and unique.

TECHNICAL CORRECTIONS

[p.7 | L26] Should read as 'true positive'

---

## Referee Comment (RC2) · Ben Mirus (Referee) · 25 Apr 2019

Summary:

This study examines the value of different information content in defining and optimizing empirical thresholds for landslide initiation using landslide and rainfall data from the Emilia Romagna Region of northern Italy. The study proposes and compares four types of thresholds, each of which include different variables and information about recent rainfall and antecedent wetness.

The study uses probabilistic thresholds and objective ROC analysis to compare the value of using 3-day recent rainfall compared to (1) using recent rainfall but also adding antecedent precipitation index (API) for a hybrid threshold, or (2) using different formulations of the API only. Although different antecedent wetness indexes could be used, the study relies only on the Antecedent Precipitation Index (API), which the authors actually show is a relatively poor indicator of soil moisture conditions. Overall the primary findings are not surprising, particularly in the context of prior research. Results show that accounting for both the triggering rainfall and the antecedent wetness (based on the API) improves threshold performance, largely by reducing false alarms. They also show that better representation of soil moisture with the API makes a substantial difference in threshold performances.

Assessment:

The issue of improving the reliability and performance of landslide initiation thresholds is an important one, and recent research has emphasized the importance of considering not only the rainfall triggering, but also the antecedent wetness conditions in landslide warning systems. The analyses presented in the manuscript are appropriate to address this issue, and the conclusions are supported by the results. The study provides a useful framework for evaluating information content, and I particularly appreciated how the figures could be used to identify when and under what conditions different types of information improved the various thresholds (e.g., API helps reduce false alarms during dry periods). The figures are clear and informative, and the manuscript is largely well written. There are some typos and awkward phrasing could be improved through additional English language editing, but these do not impede comprehension of the study or the contributions.

The primary strengths of the study are the parsimonious and systematic comparison of the role of different information content in landslide threshold performance, and the further reinforcement of the role of antecedent wetness in landslide warning systems. These nicely complement prior work on the topic. The primary weaknesses are the overly simplified and inaccurate representation of soil moisture with the API, and the lack of comprehensive consideration of different timescales for the various variables. It is worth noting that these deficiencies have been addressed to some extent by prior

studies.

As the previous reviewer also notes, the conclusions could be more specific and unique. A primary concern is the degree to which the study presents a novel and transformative contribution to the field of Hydrology and Earth Systems Science. I do believe that some readers of HESS will be interested in the topic and it would add to a growing body of literature on hydro-meteorological thresholds for landslide initiation. It could be debated whether the study is more appropriate as a technical note than a more comprehensive research article. I have noted the most pressing considerations related to this concern in the general and specific comments below.

General Comments:

I agree strongly with the background information and justification of the study, but the objectives could be clarified as they raise some questions. The authors aim to investigate two issues:

1) The role of antecedent wetness information in landslide threshold definition, which seems to be the focus of their prior work (now published in Journal of Hydrology, https://doi.org/10.1016/j.jhydrol.2019.04.062).

Although there are some differences between the new study and the authors' prior work, there are some notable similarities (e.g., overall topic, study area, data used, techniques for analysis). Thus, before considered for publication, the new work should be revised to include reference to their previous work. Specifically, the authors must provide context for how this new work goes beyond their prior contributions. One notable difference, that could be highlighted, is the framework used in the present study to evaluate the value of different types of information content in landslide thresholds.

2) Whether or not it's necessary to explicitly consider antecedent wetness, or if it's acceptable to use only the recent rainfall condition instead.

However, in the approach they use, the proxies for antecedent wetness (e.g. API)

are calculated with rainfall and temperature data only. So essentially, this study is merely comparing whether it is worthwhile to take antecedent rainfall and somehow transform it into a wetness index before developing landslide thresholds. It seems that this has already been addressed in prior studies references in the introduction (e.g., Glade, 2000; Godt et al., 2006). So even though the study uses measured soil moisture (at 10cm depth) to calibrate the recession parameter for the API, it is still a calculation with rainfall only, which is limiting. The manuscript cites our recent paper (Mirus et al., 2018a) in which we used actual soil moisture data and found similar improvements, but it does not recognize the follow up publication (Mirus et al., 2018b; https://doi.org/10.3390/w10091274) in which we further evaluated the appropriate timescales of antecedent saturation vs. recent rainfall. As such, the discussion should better recognize the limitations of the API approach in the context of other contributions in the literature (see specific comments below).

Specific Comments:

P3.L10 – In our more recent paper (Mirus et al., 2018b), we used ROC characteristics to evaluate different durations of antecedent saturation vs. recent rainfall for landslide thresholds, as well as to illustrate the impact of different choices in ROC skill metrics for hydro-meteorological threshold optimization. This is worth noting in the introduction.

P6.L15-16 – This is more or less the thresholds we identified in the aforementioned paper in Water (Mirus et al., 2018b). So it is interesting the timescales are similar.

P7.L30 – typo... should be "no landslide" not "on landslide" occur.

P8.E3&E4 – Should mention that HR and FAR are more commonly referred to as the TP_rate and FP_rate.

P8.L8 – Is this the same as the optimal point criteria (often referred to as the radial distance)?

P8.L10 – This is confusing, why would you restrict the value of HR? In ideal circumstances HR should reach unity. Do you mean that there are multiple threshold values with HR=1, but rather than minimize the Euclidean distance, you do not allow HR<1.0 and the optimization focuses on reducing FAR instead?

P10.L7-8 – Is a contingency a "null" event (i.e. day with no landslide)?

P10.L14 – Maybe not that unexpected, since the comparison between soil moisture and API is fairly poor (Figure 2).

P10.L20-26 – These findings seem consistent with Figure 7 in Zhao et al. (Journal of Hydrology, 2019). Consider discussing the differences and similarities with your prior work.

P13.L6-7 – Assuming this improvement from APIv1 to v2 mostly reflects the better representation of soil moisture (Figure 2), this suggests that even better representation of soil moisture than either API version would be even better for threshold performance. Thus, one should recommend using a better model (e.g. Godt et al., 2006), which accounts for monthly variations in ET and an exponential decline to reflect faster drainage during wetter conditions. Or even more appropriate would be to use measured soil moisture or a better model of soil moisture (Mirus et al., 2018a,b). P13.L21-22 – What do you mean by a physical-based approach? Consider providing references tha account for either the seasonality or the antecedent wetness explicitly (e.g., Napolitano et al., 2015, https://doi.org/10.1007/s10346-015-0647-5; Thomas et al., 2018, https://doi.org/10.1029/2018GL079662). Also, consider revising to "physics-based" or "physically based" rather than "physical-based."

P13.L30 – yes, see suggestions to cite in above comments (L6-7).

P14.L1-4 – Indeed, in our more recent paper (Mirus et al., 2018b) we explored a wide range of timescales and still found that 3 days does work quite well for different cities in the Pacific Northwest of the United States. Of course, different regions should expect different durations of recent rainfall to correlate with shallow landslide occurrence,

which is an important point to mention.

P14.L7-8. This was also shown by Godt et al. (2006) with a better model and by Mirus et al. (2018a,b) using actual measured soil moisture.

P14.L16-24 – Although I agree with these conclusions, they do not represent a particularly novel or unexpected finding in the context of prior published work (see references list and papers cited in this review). As such, perhaps the paper is more suitable as a technical note, than as a research paper.

P18.T2 – Not sure this table is strictly necessary or beneficial.

P19.T3&T4 – need to provide key for terms in headings, especially d, is that Euclidean distance?

P21.F2 – It seems that API in both cases is a very bad predictor of measured soil moisture. Why not use the actual measured soil moisture as we did in other studies (Mirus et al., 2018a,b)?

---

## Author Comment (AC1) · 3 May 2019

Dear Dr. Max Oke Kluger,

We are grateful for your careful reviews and constructive comments, which is very useful for the improvement of our manuscript.

All the comments are addressed point by point, and the changes are tracked in the marked manuscript. It is worth noting some revisions requested by the other reviewer are also included in the revised manuscript. It is believed that all necessary changes are made to address every point of the concerns.

You can find the response to comments, the marked and clean version of the manuscript in the supplement file. If any further information is needed, please don't

hesitate to contact us.

Yours Sincerely

Binru Zhao

Water and Environmental Management Research Centre

Department of Civil Engineering

University of Bristol, Bristol, BS8 1US, UK

Email: bz17336@bristol.ac.uk

Please also note the supplement to this comment:
https://www.hydrol-earth-syst-sci-discuss.net/hess-2019-150/hess-2019-150-AC1-supplement.zip

---

## Author Comment (AC2) · 3 May 2019

Dear Dr. Ben Mirus,

We would like to thank you for the careful reviews and constructive comments, which really lead to an improvement of our manuscript. We are especially grateful for reminding us of the publication (Mirus et al.,2018b; https://doi.org/10.3390/w10091274). We are very excited to find there are some similar methods used in the threshold definition, like the hybrid threshold. Some inspirations have been taken from your work to improve our manuscript, like Figure 4 in the revised manuscript.

All the comments are addressed point by point, and the changes are tracked in the marked manuscript. It is worth noting some revisions requested by the other reviewer

are also included in the revised manuscript. It is believed that all necessary changes are made to address every point of the concerns.

You can find the response to comments, the marked and clean version of the manuscript in the supplement file. If any further information is needed, please don't hesitate to contact us.

Yours Sincerely

Binru Zhao

Water and Environmental Management Research Centre

Department of Civil Engineering

University of Bristol, Bristol, BS8 1US, UK

Email: bz17336@bristol.ac.uk

Please also note the supplement to this comment:
https://www.hydrol-earth-syst-sci-discuss.net/hess-2019-150/hess-2019-150-AC2-supplement.zip